# OpenReview forum: "WebGuard: Building a Generalizable Guardrail for Web Agents"
_ICLR.cc/2026/Conference — ICLR 2026 Conference Withdrawn Submission_

### Official Review · Reviewer_TD3t · 2025-10-28

**Soundness:** 3
**Presentation:** 3
**Contribution:** 2
**Rating:** 6
**Confidence:** 4

**Summary:**

This paper introduces  WebGuard, a new dataset and benchmark for developing guardrails to adress the challenge of assessing the risk level of actions taken by agents.
The main contributions are:
- WebGuard Dataset: A new, large-scale dataset of 4,939 human-annotated actions from 193 real-world websites (including long-tail) across 22 domains. Actions are categorized by a novel three-tier risk schema (SAFE, LOW, HIGH).
- Benchmark Evaluation: A rigorous benchmark of frontier LLMs, demonstrating their poor zero-shot performance (e.g., <60 %accuracy and <60% recall for HIGH-risk actions).
- Fine-tuned Models: A demonstration that task-specific fine-tuning on WebGuard (e.g., with Qwen2.5-VL-7B) substantially boosts performance, with accuracy rising from $\approx$37% to $\approx$80% and HIGH-risk recall from $\approx$20% to $\approx$76%.

The central claim is that current LLMs are unsafe for this task out-of-the-box, but specialized fine-tuning on a dedicated, real-world dataset offers a promising solution, but is not yet deployment-ready.

**Strengths:**

1. Tackles a timely, important, and practical problem at the action-level. Having reliable guardrails and understanding of the impact of actions is a real bottleneck for deploying web agents in the wild safely.
2. The WebGuard dataset is a contribution. Its value comes from its scale, use of real-world websites (not simulations), domain diversity, inclusion of long-tail sites, and a well-defined annotation schema.
3. The use of four distinct generalization test splits (Long-Tail, Cross-Domain, Cross-Website, Cross-Action) is a strength, providing nuanced insights into model limitations.
4. The paper delivers clear, valuable takeaways: (a) task-specific fine-tuning (even with smaller models) drastically outperforms large zero-shot models; (b) text-only (A11y tree) inputs are better for zero-shot, but multimodal inputs are superior after fine-tuning; and (c) "reasoning models" can show lower HIGH-risk recall despite higher accuracy.
5. The paper is well-written and refreshingly transparent about the limitations of its own models, correctly stating they are not yet reliable enough for high-stakes deployment.

**Weaknesses:**

1. The paper's core contribution is a dataset and benchmark. The "guardrail construction" uses standard prompting and supervised fine-tuning with no big methodological innovation.
2. I see two concerns with the dataset methodology. First, there is no reported inter-annotator agreement (IAA): The paper reports no IAA metrics (e.g., Fleiss' Kappa), which is a standard requirement for verifying the robustness and lack of ambiguity in a new annotation schema. Second, "Default SAFE" Labeling: The strategy of labeling unannotated actions as SAFE by default could introduce bias and noise into the SAFE class if actions can be performed without having an apparent effect on the state of the page -e.g. making an API call.
3. The study fails to report on false-positive rates, which are key for agent usability as a guardrail that flags too frequently is useless. The models are also not evaluated on full, dynamic agent trajectories.
4. Key training hyperparameters are not specified, hindering full reproducibility.

**Questions:**

- Can you clarify what percentage of the 1564 SAFE samples were labeled by the "default" method versus being explicitly annotated as SAFE by a human? I would also like to know what the exact action space is, as this comment and related weakness (w2) might not make sense if no action can be performed without having and effect on the observed web page.
- Were any Inter-Annotator Agreement (IAA) scores calculated during annotation or review? If so, I think it would be useful to include them. If not, I think it should be explained and justified.
- The lack of technical novelty is a concern. Can the authors better motivate why this dataset and benchmark represents a novelty?
- The finding that "reasoning models" have lower $Recall_H$ is counter-intuitive. Were these models prompted differently? Can you provide a deeper error analysis for this failure mode?
- The zero-shot performance of Qwen2.5-VL-7B seems low. Is this a model limitation or a result of suboptimal prompting?
- The error analysis suggests the SAFE/LOW boundary is fuzzy Have you considered that the 3-tier schema itself may be too ambiguous, and another with more criteria (for example reversibility, scope, risk-type) might be more robust?

---

### Official Review · Reviewer_xLjb · 2025-10-31

**Soundness:** 3
**Presentation:** 3
**Contribution:** 2
**Rating:** 2
**Confidence:** 3

**Summary:**

This work introduces WebGuard, an action-level web interaction dataset that includes 193 real websites and >4k human-annotated actions. Based on the underlying behavior of each action, the dataset is labeled using a three-tier risk schema. The collected actions encompass multi-modal data types (e.g., screenshots, metadata, actions, etc.). Using this data, the authors fine-tune Qwen2.5-VL and build a guardrail model, demonstrating an improved recall rate for high-risk actions.

**Strengths:**

- The proposed dataset was collected from a diverse range of websites, and through cross-verified annotation, the authors evaluated each action not simply as risky or not risky but according to three levels of risk.

- In the experimental setup, rather than using a simple train/validation split, the evaluation is carefully designed to assess how robustly the fine-tuned model performs on out-of-distribution (OOD) data, such as long-tail and cross-domain scenarios.

- The authors made efforts to include refined data from a wide variety of domains and webpages in the dataset.

**Weaknesses:**

- This dataset need to be collected manually by humans, as it relies heavily on human decision-making.


- Although the authors attempted to reduce variance through cross-checking among multiple human experts, the "intent" and "outcome" of web actions cannot be automatically determined, and assessing whether an action is reversible requires subjective, experience-based judgment, which can lead to inconsistent labeling across individuals.


- In reality, the fine-tuned VLM may not have developed a deeper understanding of safety guardrails, but rather memorized the scoring rubric that the authors defined for high, low, and safe. For example, even humans would find it difficult to decide whether an action like logging in should be classified as low risk without knowing the exact rubric.

**Questions:**

- How can we judge that the fine-tuned VLM has developed a genuine understanding of safety guardrails, rather than simply memorizing the scoring rubric that defines what counts as high, low, or safe?

- Is there any other axis of the tiers in the safety we could think about, rather than just high or low?

- If the baseline VLM is provided with sufficient examples and reasoning evidence for each safety level, does its performance improve?

---

### Official Review · Reviewer_vXsr · 2025-11-01

**Soundness:** 3
**Presentation:** 3
**Contribution:** 3
**Rating:** 4
**Confidence:** 4

**Summary:**

The paper introduces WebGuard, an action‑level dataset (4939 human‑annotated actions from 193 real‑world websites spanning 22 domains) that labels each web interaction with a three‑tier risk schema (Safe / Low / High).  The authors use this to benchmark zero‑shot LLMs and to fine‑tune multimodal Qwen2.5‑VL models, demonstrating large gains (e.g., 80% accuracy and 76% high-risk recall), though this still remains insufficient for safe real-world deployment. WebGuard represents a valuable and key first step towards building reliable safety measures for web agents. While this is a good contribution, my score is conditional upon the authors addressing the concerns raised in my review.

**Strengths:**

1. The paper introduces what the authors claim is the first comprehensive action-level dataset for web agent guardrails, which uniquely includes 4,939 annotations from real-world sources and covers even the often-overlooked long-tail websites.
2. The three-stage data curation process (website selection, action collection, and a multi-reviewer annotation review) indicates that the dataset is likely to be high-quality and reliable.
3. The paper's experimental setup is comprehensive and well-designed. The authors benchmark a wide variety of frontier open source and closed source models (including GPT-4o, Claude 3.7 Sonnet, O3, Qwen2.5 series, etc.), which clearly establishes the difficulty of the task and the limitations of current out-of-the-box models

**Weaknesses:**

1. In the paper the long-tail split has only 143 actions from 15 websites. In my opinion, the split is too small for a reliable evaluation of generalization to underrepresented sites.
2. No inter-annotator agreement metrics were reported, raising questions about annotation reliability and schema clarity. How were disagreements resolved?
3. The current method of manually labelling data works for now, but it will be a bottleneck for creating the much larger datasets required in the future. Can authors suggest some strategies that could accelerate data collection while maintaining quality?
4. Table 2 lacks confidence intervals or significance tests, making it unclear whether performance differences between models are statistically meaningful, especially on small splits like long-tail tasks (143 actions).

**Questions:**

1. What was the agreement rate among annotators? How were disagreements resolved? Were any systematic disagreements observed for borderline actions?
2. Did you explore confidence thresholds or calibration techniques to reduce false negatives for HIGH‑risk actions?
3. Have you tested the fine‑tuned guardrails on completely unseen, newly launched websites (post‑release) to assess temporal robustness?
4. How much does the inclusion of the full webpage screenshot vs. only the A11y tree improve performance after fine‑tuning?
5. How many HIGH‑risk actions appear in each test split?  Did you employ any loss weighting or sampling strategies during fine‑tuning?
6. How do computational costs affect real-time agent operation? What latency do these guardrails introduce? How do they integrate with existing agent architectures?

Minor Suggestions:

To enhance readability and facilitate quick comparison across models, I recommend bolding the best-performing values in each column of Table 2.
Page 1, Line 024: It seems "lagging" is a typo for "flagging"?
The error analysis on Page 8 (Line 427) and Page 9 (Line 432) refers to "Figure D" and "Figure E". However, the figures in the appendix are numbered (Figure 4, 5). These references should be updated to point to the correct figure numbers.

---

### Official Review · Reviewer_Vvi7 · 2025-11-01

**Soundness:** 2
**Presentation:** 3
**Contribution:** 1
**Rating:** 2
**Confidence:** 3

**Summary:**

This paper introduces `WebGuard`, a new dataset and benchmark for evaluating the action-level safety of autonomous web agents. The authors argue that existing benchmarks focus on task-level success or policy adherence, neglecting the inherent risk of individual actions (e..g, "delete account"). The paper's contributions are threefold: 1) The `WebGuard` dataset, containing 4,939 human-annotated web actions from 193 websites; 2) A "novel three-tier risk schema" ($SAFE$, $LOW$, $HIGH$) for classifying these actions; and 3) Baseline experiments showing that frontier LLMs perform poorly on this task ( <60% HIGH-risk recall), while a fine-tuned Qwen2.5VL-7B model improves performance (80% accuracy, 76% HIGH-risk recall). The authors posit this as a critical step toward building reliable agent guardrails.

**Strengths:**

1. **Targeted an Important and Under-Studied Problem:** The paper correctly identifies the gap in agent safety. It shifts the focus from task-level policy compliance (like in `ST-WebAgentBench`) or malicious-intent red-teaming (like in `BrowserART`) to the more granular, inherent risk of *atomic actions*. This is a timely and promising research direction.
2. **Valuable Dataset Artifact:** The `WebGuard` dataset is a substantial contribution. With 4,939 annotated actions across 193 diverse websites, it provides a high-utility resource for the community to benchmark future safety models.
3. **Strong Negative Result:** The paper clearly demonstrates that out-of-the-box frontier LLMs are dangerously deficient in this task ( <60% recall on high-risk actions). This finding is valuable as it proves that this capability is not an emergent property of general-purpose models and requires dedicated research.

**Weaknesses:**

1. **Contribution is Heavily Engineering-Focused:** The paper's core contribution is a dataset, and it lacks the methodological novelty typically expected at ICLR.
   - The conceptual framework of an "action-impact" dataset is not new; "Interaction to Impact" recently established this paradigm for mobile UIs. This work is a valuable *port* to the web domain but not a new research *concept*.
   - The proposed $SAFE/LOW/HIGH$ risk schema is a standard, pragmatic simplification used in operational risk assessment, not a novel research-driven taxonomy.
   - The proposed model (SFT on Qwen2.5VL-7B) is a simple baseline, not a novel method. This is especially apparent when compared to concurrent, reasoning-based frameworks like `GuardAgent`.
2. **Performance Gap Demonstrates Solution is Not Practically Usable:** As the authors note, the model's performance "falls short," but the severity of this is understated.
   - The paper's best model achieves only **~76% recall on HIGH-risk actions**.
   - For a safety-critical system, this is not a small gap; it is a critical failure. It means the guardrail **misses 1 in 4 catastrophic actions** (e.g., "delete account," "confirm purchase").
   - While the paper successfully *benchmarks* the problem, the proposed solution is far from being practically usable and underscores the deep difficulty of the challenge, which the paper's method does not solve.

**Questions:**

1. Could the authors better delineate their conceptual novelty from "Interaction to Impact", which already introduced the paradigm of action-level impact assessment for mobile UIs?
2. The SFT baseline achieves only 76% recall on high-risk actions. Why did the authors not explore more complex, verifiable methods, such as a reasoning or code-generation framework (e.g., `GuardAgent`), which might offer more robust checks than an end-to-end classifier?
3. What was the reasoning for choosing a simple 3-tier $SAFE/LOW/HIGH$ schema over a more descriptive, multi-dimensional taxonomy of impact, as explored by "Interaction to Impact"? Does this simplification risk over-generalizing different *types* of high-risk actions (e.g., financial vs. privacy vs. destructive)?

---

### Note · Authors · 2025-11-29

**Comment:**

We thank all reviewers for their time and engagement with our submission. After carefully reviewing the feedback, we have decided to withdraw this paper. While we appreciate the constructive points raised. We wish to respectfully address several concerns about the reviews before withdrawing.

### On model performance ('The fine-tuned model is not deployment-ready')
It is mentioned that our best model is not "deployment-ready." We respectfully suggest this misunderstands our contribution. WebGuard is the first work to formally define the action-level guardrail problem for web agents. The performance gap we report is the motivation for this research, demonstrating that even frontier LLMs fail at this task establishes web agent safety as a genuinely unsolved problem requiring dedicated attention.

### Lack of method contributions
Several reviews note the lack of methodological novelty, and the contribution is "heavily engineering-focused". WebGuard is positioned as a dataset and benchmark contribution, not a methods paper. This work aims to establish a benchmark and a formulation to approach the safety problem of web agents.

We thank the reviewers again for their time and thorough engagement with our work.

**Withdrawal Confirmation:**

I have read and agree with the venue's withdrawal policy on behalf of myself and my co-authors.